Over-expression of CcMYB24, encoding a R2R3-MYB transcription factor from a high-leaf-number mutant of Cymbidium, increases the number of leaves in Arabidopsis

Li Gengyun
Cheng Longjie
Li Zhilin
Zhao Yiran
Wang Yuying ynauhort@163.com
College of Landscape and Horticulture, Yunnan Agricultural University , Kunming , China
Irfan Mohammad
Electronic publication date: 2023 May 31
Publication date: 2023
Volume: 11
Electronic Location ID: e15490
Received 2023 Feb 1; Accepted 2023 May 10
Copyright: © 2023 Li et al.
Copyright year: 2023
Copyright holder: Li et al.
License: This is an open access article distributed under the terms of the Creative Commons Attribution License, which permits unrestricted use, distribution, reproduction and adaptation in any medium and for any purpose provided that it is properly attributed. For attribution, the original author(s), title, publication source (PeerJ) and either DOI or URL of the article must be cited.
License URL: https://creativecommons.org/licenses/by/4.0/

Keywords: CcMYB24, Cymbidium, Leaf number, Ornamental foliage plants, MYB transcription factor, Mutant

Funding: Provincial Innovation Team of Yunnan Province of China 202105AE160012 Yunnan Province Basic Research Project of China 2018FB070, 202001AT070101 and 202301AT070494 National Key R&D Program of China 2022YFF1302402 This study was financially supported by the Provincial Innovation Team of Yunnan Province of China (Grant No. 202105AE160012), Yunnan Fundamental Research Projects (Grant No. 2018FB070, 202001AT070101, and 202301AT070494) and National Key R&D Program of China (Grant No. 2022YFF1302402). The funders had no role in study design, data collection and analysis, decision to publish, or preparation of the manuscript.

==============================
Ornamental foliage plants have long been cultivated for their attractive leaves. Variation in leaf traits of ornamental foliage plants is one of the goals in breeding. MYB transcription factors regulate many aspects of leaf development, and thus influence morphological traits of leaves. However, little is known about the function of MYB transcription factors in leaf development of Cymbidium, one of the most economically important ornamental plants in the world. In the present study, a MYB transcription factor, CcMYB24, was identified and the corresponding gene cloned from a new orchid mutant, TRIR-2, which produces more leaves than control plants. The CcMYB24 showed a higher expression level in ‘TRIR-2’ than in control plants, and the protein was located in the nucleus. The sequence of CcMYB24 showed a high similarity with RAX2-like genes which belong to the R2R3-MYB gene family in other Cymbidium plants. Overexpression of CcMYB24 resulted in a phenotype with an increased number of leaves, elevated chlorophyll content, and decreased contents of carotenoids and flavonoids in Arabidopsis. These results provide functional evidence for the role of CcMYB24 in promoting the production of leaves in ‘TRIR-2’. Understanding the role of CcMYB24 in Cymbidium will be beneficial for the molecular breeding of ornamental foliage plants.

Introduction

As an important organ for photosynthesis, leaves are usually exposed to a variety of ecological and environmental factors such as light, temperature and herbivores, and play a critical role in the growth and development of plants. The morphology of leaf and the pattern of shoot branching largely determine the growth habit of seed plants. In recent years, several genes involved in leaf development have been identified and characterized in plants. Most of them are transcription factors (TFs) that form a complex regulatory network to control the development of various leaf traits (Du, Guan & Jiao, 2018; Wang, Kong & Zhou, 2021). A deeper understanding of the molecular basis of leaf development is needed, not only to improve crop production but also to deepen our overall comprehension of plant biology.

The v-myb avian myeloblastosis viral oncogene homolog (MYB) genes represent one of the largest gene superfamilies encoding TFs in eukaryotes (Rosinski & Atchley, 1998; Jiang & Rao, 2020). A number of studies has shown that the MYB gene family underwent sizable functional diversification during evolution, to form different subfamilies or groups (Jiang & Rao, 2020; Li et al., 2020; Wu et al., 2022). Based on the number of conserved MYB domain at the N-terminus, which contains one to four imperfect repeats (R) associated with DNA-binding and protein–protein interactions, MYB TFs can be classified into four different groups, namely the 1R, R2R3, 3R, and 4R classes, which contain one to four repeats, respectively (Dai et al., 2020; Wang et al., 2020). Among them, the R2R3-MYB proteins form the largest subfamily of MYB TFs, although they only exist in terrestrial plants (Dai et al., 2020; Wang et al., 2020). The R2R3-MYB proteins can act as transcriptional activators as well as repressors in plants, and most of them are activators which function by binding cis-elements in gene promoters to recruit transcriptional machinery and activate gene expression (Wu et al., 2022). A series of biological functions of R2R3-MYBs have been verified in model plants and non-model crops in recent years with the development of approaches which focus on gene functional verification. In summary, the R2R3-MYBs have been shown to be involved in regulating specialized metabolic pathways, such as anthocyanin and phenylpropanoid biosynthesis pathways, response to biotic and abiotic stresses, such as fungal pathogens and drought or salt stresses, participation in plant development and cell differentiation, and responses to hormone signaling (Stracke, Werber & Weisshaar, 2001; Wang, Niu & Zheng, 2021; Ambawat et al., 2013; Wu et al., 2022). Therefore, MYBs are a class of important TFs necessary for the basic life cycle of plants, and more research is needed to determine their functions in basic developmental processes.

Leaf differentiation and morphogenesis are highly flexible processes that are regulated by MYB genes. Previous studies had shown that the MYB domain proteins play a key role in leaf polarity establishment, organ initiation and the subsequent developmental processes (Müller, Schmitz & Theres, 2006; Ge & Chen, 2014). Three RAX genes in Arabidopsis thaliana, RAX1, RAX 2 and RAX 3, are members of the R2R3-Myb gene family. These genes control the formation of axillary meristems at different developmental times (Müller, Schmitz & Theres, 2006). There is also some evidence showing that sub- or neo-functionalization of A. thaliana RAX-1 orthologs regulates the architecture of leaves and shoots in Solanum lycopersicum (Busch et al., 2011). Some MYB genes, such as RS2 and PHAN, participate in leaf development by negatively regulating KNOX genes that determine the acquisition of founder cell identity in the meristem during leaf initiation (Tsiantis et al., 1999). Another case involves the MYB gene ARP, which not only plays an important role in establishment of proper polarities along the adaxial-abaxial, proximodistal, and medial-lateral axes, but also participates in the regulation of leaf petiole identity, as indicated by evidence showing that suppression of ARP expression led to a series of abnormal leaf developmental processes in Medicago truncatula and Nicotiana benthamiana (Huang et al., 2013; Ge & Chen, 2014). Such findings indicate that MYB transcription factors may influence leaf traits in multiple pathways. With the development of gene function verification technology, it can be expected that more functions of MYB transcription factors in relation to the regulation of leaf traits will be revealed in different plant taxa in the future.

Cymbidium is one of the most economically important genera of the Orchidaceae and is highly popular around the world for its beautiful flowers and leaf variegation (Jiang et al., 2022). A mutagenesis-based approach is the main way to produce various flower and leaf color mutants in different Cymbidium species (Jiang et al., 2022). Leaf traits, such as color, variegation patterning, shape and number of leaves, are important in determining the quality of ornamental foliage plants. In China, the variation in leaf traits in ornamental foliage plants has been called ‘leaf art’, and novel variation in leaf traits is one of the goals in ornamental plant breeding. In our previous study, we obtained a new orchid mutant, temporarily named ‘TRIR-2’, through radiation mutation breeding on a hybrid between two Cymbidium species (Su et al., 2016). A comparative cytological study between the non-mutant Cymbidium inter-specific hybrid ‘TRIR-1’ and ‘TRIR-2’, which was not induced by irradiation, revealed significant differences. The study found that ‘TRIR-2’ had fewer chloroplasts, mitochondria, and other organelles, and some leaf cells of ‘TRIR-2’ contained only minimal cytoplasm. Consequently, the leaves of ‘TRIR-2’ appeared albino, with green color only visible at the edges and tips (Su et al., 2016). Furthermore, it was obvious that ‘TRIR-2’ generated more leaves than ‘TRIR-1’. According to our unpublished data, the result of RNA sequencing had revealed that expression of a MYB gene CcMYB24 was significantly upregulated in ‘TRIR-2’, compared with ‘TRIR-1’ as the control, indicating that this gene may contribute to the morphological differences in ‘TRIR-2’.

In the present study, we describe the cloning and characterization of CcMYB24 from the ornamental foliage mutant line ‘TRIR-2’. The sequence analysis and protein structure prediction showed that CcMYB24 belongs to the R2R3-MYB subfamily. Overexpression of CcMYB24 in transgenic Arabidopsis seedlings significantly increased the number of leaves. These results indicate that this gene plays a positive role in increasing the number of leaves in ‘TRIR-2’, providing a basis for exploring the molecular mechanism of CcMYB24 in leaf development, and identifying a potential gene target for the molecular breeding of new cultivars in ornamental foliage plants.

Materials and Methods

Plant materials and growth conditions

The mutant plant material ‘TRIR-2’ used in this study was produced by 60Co γ-irradiation on ‘TRIR-1’, the latter being the hybrid offspring of two Cymbidium species, Cymbidium tracyanum × Cymbidium iridioides. In brief, the two Cymbidium species, C. tracyanum and C. iridioides, were first cross-fertilized to generate ‘TRIR-1’, an orchid hybrid with a stable phenotype. Then, ‘TRIR-1’ was exposed to 60Co γ-rays to induce mutations. Several individuals expressing variation in leaf traits were identified. After subculture and screening for 8 years, we isolated the stable mutant named ‘TRIR-2’. ‘TRIR-2’ showed distinct phenotypic variation in leaf traits compared with the parental hybrid individuals. Pure-breeding seedlings of ‘TRIR-1’ and ‘TRIR-2’ were grown in a climate-controlled incubator (MGC-350HP-2; Yiheng Technical Co., Ltd, Shanghai, China) under long-day conditions (16 h light/8 h dark) at 22 °C in the growth chamber. Several traits were measured and compared between ‘TRIR-2’ and non-irradiated ‘TRIR-1’ individuals grown under the same conditions, including plant height (the distance between the highest leaf tip and soil), number of leaves, length of leaves (the distance from tip to base of mature leaf) and breadth of leaves (the width of the widest part of mature leaf). Plant material cultivation was performed in the College of Horticulture and Landscape, Yunnan Agricultural University, Kunming, China.

Cloning and sequence analysis of CcMYB24

The expressed sequence tag of CcMYB24 (File S1) was aligned to NCBI using BLAST. As a result, the RAX2-like gene (query coverage: 100%; percentage identity: 96.81%) of Cymbidium ensifolium showed the greatest sequence similarity and was used to design degenerate primers to obtain the full-length sequence of CcMYB24 in ‘TRIR-1’ and ‘TRIR-2’. The sequences of primers were listed in Table S1. The sequence of ‘TRIR-1’ and ‘TRIR-2’ was aligned to identify their sequence differences by using Multiple Sequence Alignment function in DNAMAN 8.0. Then, we used ORFfinder (https://www.ncbi.nlm.nih.gov/orffinder) to obtain the coding sequence (CDS) and amino acid sequence of the corresponding protein, and aligned the latter to NCBI using BlastP to screen for homologs. The homology was compared using DNAMAN 8.0 software. Additionally, phylogenetic trees were constructed based on full-length amino acid sequences by the Maximum Likelihood method in MEGA 7.0 software, with bootstraps set to 1,000.

Prediction of physicochemical properties and structure of CcMYB24 protein

The conservation of CcMYB24 domains was examined using NCBI Conserved Domains (www.ncbi.nlm.nih.gov/Structure/cdd/wrpsb.cgi) (Yang et al., 2020). The online tool ExPASy-ProtParam (https://web.expasy.org/protparam/) and ExPASy Protscale analysis program (https://web.expasy.org/protscale/) was used to estimate the physicochemical parameters (including molecular weight, theoretical pI (isoelectric point), instability index, aliphatic index and grand average of hydropathy (GRAVY)), hydrophobicity and hydrophilicity values of the CcMYB24 protein, by providing sequence information and analyzing in the website following the user guide (Gasteiger et al., 2003). The TMHMM-2.0 (https://services.healthtech.dtu.dk/service.php?TMHMM-2.0) tool was adopted to predict the transmembrane helical region, which gives some statistics and a list of the location of the predicted transmembrane helices and the predicted location of the intervening loop regions. To predict the presence of a signal peptide, SignalP-5.0 software (https://services.healthtech.dtu.dk/service.php?SignalP-5.0) was used by uploading protein sequence and analyzing by online tools. The SOPMA software (http://npsa-pbil.ibcp.fr/cgi-bin/npsa_automat.pl?page=npsa_sopma.html) and SWISS-MODEL (http://www.expasy.ch/swissmod/SWISS-MODEL.html) software were used to predict the secondary structure and tertiary crystal structure following the online guideline of programs, respectively (Geourjon & Deléage, 1995; Arnold et al., 2006).

Subcellular localization

The subcellular localization of CcMYB24 was initially predicted using the pSORT web server (http://www.genscript.com/psort/wolf_psort.html). Then, the coding sequence of CcMYB24 was amplified by using the primers listed in Table S1 and cloned into the vector pCAMBIA1300-35S-N-GFP, which expresses the green fluorescent protein (GFP), using EcoR. and SalI restriction enzymes. The recombinant vector was introduced into competent Escherichia coli DH5-Alpha cells and then transformed into Agrobacterium tumefaciens EHA105. The Agrobacterium was injected into the leaves of four-week-old seedlings of Nicotiana benthamiana. Following the subsequent transient gene expression, subcellular localization of the CcMYB24-GFP fusion protein was observed using confocal laser scanning microscopy (LSM710; Carl Zeiss, Oberkochen, Germany) for 72 h after the injection (Olympus, Tokyo, Japan).

Vector construction and transformation of CcMYB24 in Arabidopsis

DNA fragments containing the full-length coding sequence of CcMYB24 were amplified from the DNA of ‘TRIR-2’ using gene-specific primers (Table S1) and then cloned into the pCAMBIA1300-35S vector between SacI and BamH. restriction sites. Then, the recombinant vector was introduced into A. tumefaciens strain EHA105 and transformed into Arabidopsis wild-type (WT) Columbia ecotype (Col-0) via the floral dip method (Clough & Bent, 1998). The seeds of transformed Arabidopsis were surface sterilized with ClO2 for 8 min, rinsed thoroughly and selected for growth on selection medium containing 40 mg/L hygromycin. The Arabidopsis plants, including WT and genetically transformed individuals, were grown in incubator under long-day conditions (16 h light/8 h dark) at 23 °C during the light time and 21 °C during dark time, respectively. The presence of the gene insert was confirmed by PCR using 2×T5 DIRECT PCR Kit (Tsingke Biotech Co., Ltd., Beijing, China) with the primers listed in Table S1.

Quantification of CcMYB24 gene expression by qRT-PCR analysis

The expression of CcMYB24 gene was quantified by quantitative reverse transcription-PCR (qRT-PCR) in leaf of forty-day-old ‘TRIR-2’ and non-irradiated ‘TRIR-1’ Cymbidium seedlings, as well as in stem, leaf, and flower of genetically transformed and WT Arabidopsis. The RNAs were extracted by using the RNAprep Pure Plant Plus Kit (Tiangen, Beijing, China), following the manufacturer’s instructions. For each sample, 1.0 µg of total RNA was reverse transcribed into cDNA using a PrimeScriptTM RT reagent kit (TaKaRa, Dalian, China). The qPCR was performed using SYBR® Premix Ex Taq Kit (TaKaRa, Dalian, China) and ABI7900 Fast Real-Time PCR System (ABI, Foster City, CA, USA). The 18S rRNA gene and UBQ5 were selected as the internal reference gene for TRIR-1/2 and Arabidopsis, respectively. The expression value was calculated by using the 2−ΔΔCt method based on three biological replications for each sample. The relative expression was further calculated in Arabidopsis using log2 fold change of genetically transformed vs. WT plants (Li et al., 2017). Primers are listed in Table S1, and the Cq value for each sample was listed in Table S2.

Trait measurement in Arabidopsis

Several traits were measured and compared in genetically transformed and WT Arabidopsis. Morphological traits: number of leaves and branching were measured after 4 and 5 weeks of growth, respectively; physiological traits were measured after 4 weeks of growth: the concentration of photosynthetic pigments including chlorophyll a, chlorophyll b, total chlorophyll, and total carotenoids, were determined by the spectrophotometric method that homogenizing plant tissue in a solvent, centrifuging the homogenate, and analyzing the supernatant’s absorbance at specific wavelengths to calculate the concentration of photosynthetic pigments (Jodłowska & Latała, 2011); the soluble sugar concentration was measured by the anthrone method, which involves adding anthrone reagent to the sample and heating it in a water bath. The colored complex formed between the anthrone and the sugar is then measured using a spectrophotometer, and the concentration of soluble sugars is calculated based on a standard curve (Clegg, 1956); the soluble protein concentration was determined by the Coomassie Brilliant Blue G-250 (Snyder & Desborough, 1978), which involves adding the Coomassie Brilliant Blue G-250 reagent to a protein sample, allowing it to bind to the protein, and measuring the absorbance of the resulting complex at a specific wavelength using a spectrophotometer. The protein concentration is then determined by comparing the absorbance of the sample to a standard curve of known protein concentrations. The total flavonoid concentration was measured by the Al(NO3)3-NaNO2-NaOH colorimetric assay (Zheng et al., 2011), which involves the formation of a colored compound specific to flavonoids, and its absorbance is measured at a specific wavelength using a spectrophotometer.

Statistical analysis

All the data are presented as the mean ± standard error of at least six biological replicates for morphological traits and three biological replicates for physiological traits to confirm reproducibility. Statistical analyses were conducted using SPSS 22.0 (IBM, Armonk, NY, USA). Differences between two groups were calculated using Student’s t-test. Statistical significance was defined as p < 0.05 (*) or p < 0.01 (**). The raw data are listed in Tables S3–S5.

Results

Variation in morphological traits between ‘TRIR-2’ and ‘TRIR-1’

Individuals of mutant ‘TRIR-2’ showed distinct variation with respect to several morphological traits in comparison with wild-type ‘TRIR-1’. Whereas ‘TRIR-1’ has normal green leaves, the leaves of TRIR-2 are mostly white, with green color appearing only at the edges and tip of each leaf (Fig. 1A). The ‘TRIR-1’ plants were significantly taller than ‘TRIR-2’ plants of the same age (p = 0.03, Fig. 1B). Although the length of leaves was significantly longer in ‘TRIR-1’ than in ‘TRIR-2’ (p < 0.01, Fig. 1C), ‘TRIR-2’ produced significantly more leaves than TRIR-1 did (p = 0.02, Fig. 1D), and the leaves of ‘TRIR-2’ were wider than those of ‘TRIR-1’ (p = 0.04, Fig. 1E).

Figure 1 Variation in leaf traits between ‘TRIR-2’ and ‘TRIR-1’.

(A) Overall phenotype of ‘TRIR-2’ and ‘TRIR-1’. (B–E) Comparison of height (B), length of leaves (C), number of leaves (D) and breadth of leaves (E) between TRIR-2 and TRIR-1. Data are presented as means ± SE (n = 6). Statistically significant differences were represented by * (p < 0.05) or ** (p < 0.01), following Student’s t-test.

Cloning and sequence analysis of CcMYB24

The complete sequence of CcMYB24 was amplified and sequenced from ‘TRIR-1’ and ‘TRIR-2’. The full-length sequence of CcMYB24 was 1389 bp, and the length of the open reading frame (ORF) was 906 bp, encoding 301 amino acids (File S1). The amplified sequence of ‘TRIR-1’ was determined to be 861 base pairs in length. Upon sequence alignment with the corresponding sequence of ‘TRIR-2’, it was observed that ‘TRIR-1’ exhibited a segmental variation of 95 base pairs, including three gaps with a deletion of 45 base pairs, and 50 single nucleotide polymorphisms (SNPs) when compared to ‘TRIR-2’. These observed variations led to a loss of 15 amino acids and a substitution of 24 amino acids in the peptide chain of ccMYB24 in ‘TRIR-1’. The alignment results are provided in File S2. Analysis of conserved domains showed that the protein encoded by CcMYB24 had a conserved Myb DNA-binding domain (Fig. 2A). According to the results of BLAST, the ten sequences with the highest amino acid similarity to CcMYB24 were obtained, including CeRAX2-like (from Cymbidium ensifolium, 95.99% similarity), DcRAX2 (from Dendrobium catenatum, 86.15% similarity), PeRAX2-like (from Phalaenopsis equestris, 76.49% similarity), EgRAX2-like (from Elaeis guineensis, 60.80% similarity), PdMYB36 (from Phoenix dactylifera, 64.17% similarity), MeRAX2 (from Manihot esculenta, 52.51% similarity), JrRAX2-like (from Juglans regia, 51.39% similarity), QlRAX2-like (from Quercus lobata, 45.12% similarity), QsRAX2-like (from Quercus suber 50.53% similarity), NtRAX (from Nicotiana tabacum, 45.85% similarity) and AtRAX2 (from A. thaliana, 39.60% similarity). All of these genes belong to the R2R3-MYB family which includes most MYB TFs (Kelemen et al., 2015). The result of sequence alignments showed that all these sequences had conserved domains of the Myb superfamily proteins, such as the PLN03091 superfamily, REB1 superfamily, and two MYB DNA-binding domains (Fig. 2B). Phylogenetic analysis showed that the CcMYB24 protein is closely related to the CeRAX2-like protein and the DcRAX2 protein (Fig. 2C).

Figure 2 Sequence analysis of CcMYB24.

(A) The conserved domain revealed by BLASTP search of the database, using CcMYB24 as the query. (B) Multiple sequence alignment of the CcMYB24 protein and its closest homologs from different plant species. All these sequences have conserved domains of MYB superfamily proteins such as PLN03091 superfamily, REB1 superfamily, and two Myb DNA-binding domains. (C) Phylogenetic relationships of CcMYB24 and its closest homologs from different plant species. Node values are percentages of bootstraps generated, with n = 1,000 bootstrap replicates. The CcMYB24 cloned in this study is indicated in bold font.

The physicochemical properties and structure of CcMYB24 protein

The ProtParam ExPaSy program was used to analyze the amino acid sequence and chemical properties of the CcMYB24 protein. The result showed that CcMYB24 possesses the molecular formula C1457H2263N407S15, and the molecular weight was calculated to be 33,242.35 Da. The theoretical isoelectric point (pI) was 8.59, and the instability index was 63.02, indicating that the protein is unstable. The grand average value of the hydropathy index (GRAVY) was calculated as the sum of the hydropathy values of all amino acids, and the estimated value of GRAVY was −0.483 (Fig. 3A). This finding suggests that the CcMYB24 protein is likely to be largely hydrophilic. The result of the transmembrane helix region and signal peptide prediction analysis showed that neither transmembrane structure nor signal peptide could be found in the CcMYB24 protein, indicating that it is not a transmembrane protein, nor a secretory protein. The secondary structure of the protein was predicted using the SOPMA server and shown in Fig. 3B. The result showed that CcMYB24 protein was mainly composed of random coil (52.49%), alpha-helix (24.58%), extended strand (16.28%) and beta turn (6.64%). The prediction of the tertiary protein structure of CcMYB24 (Fig. 3C) by the SWISS-MODEL analysis platform showed a similar structure with a R2R3-MYB transcription factor from A. thaliana which has six alpha-helices belonging to R2 and R3 repeats (Wang et al., 2020).

Figure 3 Physicochemical characterization and structure of CcMYB24.

(A) Analysis of the hydrophobicity of the CcMYB24 protein. (B) Secondary structure prediction of the CcMYB24 protein. The alpha-helix, beta turn, random coil and extended strand are showed as lines in blue, green, purple and red, respectively. (C) Predicted 3D spatial structure of CcMYB24 protein.

Subcellular localization of CcMYB24

Most R2R3-MYB proteins are exclusively localized in the nucleus, where they function as TFs (Stracke, Werber & Weisshaar, 2001). For CcMYB24, the result of localization prediction using the pSORT web server predicted that it was located in the nucleus. We further determined the subcellular localization of CcMYB24 via transient expression of a translation fusion with GFP in tobacco leaves. The result showed that the fluorescence signal of the GFP control was localized to the cytoplasm and the nucleus in N. benthamiana leaves, whereas the CcMYB24-GFP was located exclusively in the nucleus (Fig. 4), a finding similar to that of other R2R3-MYB TFs, suggesting that CcMYB24 also functions in the nucleus as a TF.

Figure 4 Subcellular localization of CcMYB24 in Nicotiana benthamiana leaves.

GFP, green fluorescent protein fluorescence images. DIC, differential interference contras. MERGE, combined images. Scale = 50 μm.

Expression pattern of CcMYB24 in ‘TRIR-1’, ‘TRIR-2’ and transgenic Arabidopsis

The expression pattern of CcMYB24 was examined in wild-type Cymbidium ‘TRIR-1’, mutant Cymbidium ‘TRIR-2’ and transgenic Arabidopsis by using quantitative real-time PCR (qPCR). As shown in Fig. 5A, the relative expression of CcMYB24 transcripts was about 1.4 time higher in ‘TRIR-2’ than in ‘TRIR-1’, indicating that more transcripts were generated in leaves of ‘TRIR-2’ than in those of the wild-type parent. In the case of Arabidopsis, the expression level was evaluated in terms of log2 fold change between transgenic and WT individuals. The result showed that CcMYB24 expression was much higher in the stem, leaf and flower of transgenic Arabidopsis than in the corresponding organs of the WT (Fig. 5B).

Figure 5 The relative expression pattern of CcMYB24 in ‘TRIR-1’, ‘TRIR-2’, and transgenic Arabidopsis revealed by using qRT-PCR.

(A) Relative expression of CcMYB24 transcripts in ‘TRIR-1’ and ‘TRIR-2’. (B) Log2 fold change of relative expression level in different organs of genetically transformed vs. WT plants. Relative expression quantification was carried out using the 2−ΔΔCt method, with 18S rRNA and UBQ5 as the internal reference gene for ‘TRIR-1/2’ and Arabidopsis, respectively. Error bars represent mean ± SE (n = 3).

CcMYB24 induced production of more leaves when overexpressed in Arabidopsis

To investigate the function of CcMYB24 in ‘TRIR-2’, the coding region was expressed in A. thaliana Col-0. We observed some differences between the transgenic and WT individuals (Figs. 6A–6D). Despite a p-value of 0.06, statistical analysis revealed no significant difference in the number of branches between WT and transgenic individuals (Fig. 6E). The mean number of leaves in transgenic individuals was significantly higher than that in WT plants (p < 0.01, Fig. 6F).

Figure 6 Comparison of morphological traits between CcMYB24 genetically transformed and WT Arabidopsis.

(A and B) Morphological phenotype of genetically transformed (A) and WT (B) Arabidopsis at vegetative growth stage. (C and D) Morphological phenotype of genetically transformed (C) and WT (D) Arabidopsis at reproductive growth stage. (E and F) Comparison of number of branches (E) and number of leaves (F) between genetically transformed and WT Arabidopsis. Overexpression of CcMYB24 in transgenic Arabidopsis plants led to a significant increase in the number of leaves. Data are presented as means ± SE (n = 23), and the statistically significant differences are represented by ** (p < 0.01), using Student’s t-test.

CcMYB24 markedly affected the accumulation of photosynthetic pigments and total flavonoids

The influence of CcMYB24 on a number of physiological traits, namely photosynthetic pigments, total flavonoids, and soluble sugar and protein, were examined and compared between transgenic and WT Arabidopsis. For photosynthetic pigments, the results showed that chlorophyll a (p < 0.01), chlorophyll b (p = 0.02), and total chlorophyll concentrations (p < 0.01) were significantly higher in transgenic Arabidopsis (Fig. 7A), whereas total carotenoid concentration was significantly lower in transgenic Arabidopsis plants than in WT individuals (p < 0.01, Fig. 7A). The total flavonoid concentration was also significantly lower in transgenic Arabidopsis than in WT individuals (p < 0.01; Fig. 7B), but soluble sugar and protein concentrations did not differ significantly between transgenic and WT plants (Figs. 7C and 7D).

Figure 7 Comparison of concentration of (A) photosynthetic pigments, (B) total flavonoids, (C) soluble sugar, and (D) soluble protein between CcMYB24 genetically transformed (blue) and WT (red) Arabidopsis.

Overexpression of CcMYB24 causes significant changes in concentration of photosynthetic pigments and total flavonoids. Data are presented as means ± SE (n = 3). Statistically significant differences are represented by * (p < 0.05) or ** (p < 0.01), using Student’s t-test.

Discussion

Cymbidium plants enjoy a prime position in the floriculture industry. The ornamental value of Cymbidiums is conferred not only by the large and colorful flowers and extended flower longevity, but also by its foliage and floral fragrance (Jiang et al., 2022). Some Cymbidium varieties have long been cultivated as foliage plants which are grown for attractive leaves, and the combined character of leaf variegation and colorful flowers have an added advantage in the ornamental plant market. Variations in leaf traits, including color, shape and number of leaves, are vital for the quality of foliage plant. In this study, a new foliage variant of Cymbidium was generated by combining inter-specific hybridization with radiation mutation breeding. The new variant, ‘TRIR-2’, has significantly broader but shorter leaves compared with the parent, ‘TRIR-1’, and showed an increased number of leaves and a part-albino leaf phenotype, with a large area of white in the middle of the leaves with green only at the edges and tip of the leaf (Fig. 1A), making it an attractive ornamental plant.

Several researches have explored mutation in leaf development including shape development, photosynthetic mechanisms, chloroplast development and other mechanisms of genetic control on color variation of leaves. These studies mainly focused on model plants or crops such as A. thaliana (Moon et al., 2008), Zea mays (Lonosky et al., 2004), Nicotiana tabacum (Wu et al., 2020), and Oryza sativa (Zheng et al., 2019). The knowledge of genetic control of number of leaves is still limited, especially in ornamental foliage plants which commercial value strongly influenced by the number of leaves. Recent studies have shown that transcription factors control many aspects of leaf developmental processes (Liu et al., 2018; Zhao et al., 2020; Heisler et al., 2022). The MYB transcription factors are versatile in regulation of development, stress responses as well as in leaf development (Jiang & Rao, 2020; Wang, Niu & Zheng, 2021; Wu et al., 2022), but the knowledge about the role it plays in control of number of leaves is still limited.

In the present study, a R2R3-MYB gene, CcMYB24, was reported to participate in producing more leaves in ‘TRIR-2’, a high-leaf-number mutant of Cymbidium. The expression analysis showed that CcMYB24 exhibited a higher expression level in ‘TRIR-2’ than in ‘TRIR-1’. The mechanism of expression level increasing is complex and remain elusive. It may be due the mutation of upstream and/or regulatory gene, or the difference in epigenetic control between ‘TRIR-1’ and ‘TRIR-2’. More works are needed in the future to discover the reason why there were more transcripts of CcMYB24 in ‘TRIR-2’. We plan to conduct comparative transcriptome and gene co-expression network analysis between ‘TRIR-1’ and ‘TRIR-2’ and further integrated the results with the analysis of epigenetic control of CcMYB24.

According to the results of sequence similarity analysis, CcMYB24 appeared to be a RAX2-like gene which belongs to the R2R3-MYB transcription factor gene family. Many studies have proved that the RAX homolog genes play important roles in determination of axillary meristems by generating a tissue environment conducive to the establishment of meristems to control the spatial pattern of axillary meristems development (reviewed by Zhang et al., 2022), thus participate in branching and leaf development. In Arabidopsis, transcripts of RAX genes accumulated in the axils of young leaf primordia and controlled a very early step of axillary meristem initiation, with rax mutants showing new phenotypes characterized by defects in lateral bud formation in overlapping zones along the shoot axis (Müller, Schmitz & Theres, 2006). Much evidence has also been reported that RAX genes control axillary meristem formation and shoot branching in Arabidopsis by collaborating in gene modules with other transcriptional factors, such as bHLH and WRKY (Yang et al., 2012; Guo & Qin, 2016). In Chrysanthemum morifolium, CmRAX2 was expressed in the lateral branches and roots, with overexpression of CmRAX2 leading to increases in lateral branch number and plant height (Song et al., 2022). RAX genes were also be found to participate in shoot branching and leaf dissection in tomato (Busch et al., 2011; Zhang et al., 2022). Based on these evidences, it could be inferred that the CcMYB24 may be responsible for the variation in foliage traits shown by ‘TRIR-2’.

The current studies showed the regulatory role played by RAX2 genes in shoot branching and leaf development. The results of transgenic experiment in this study showed that overexpression of CcMYB24 in Arabidopsis stimulated production of more leaves (Fig. 6E) and more branches (Fig. 6F), although the latter effect was not significant (but close with p-value of 0.06), indicating that CcMYB24 may contribute to the phenotype of more leaves in ‘TRIR-2’ compared with ‘TRIR-1’. Orchid plants grow by only one or two growth patterns, namely sympodial or monopodial. Both types grow by leaves rather than obvious branches. Therefore, it can be inferred that the phenotype of more leaves in ‘TRIR-2’ may be related to the branching regulation nature of RAX2 genes. However, the factors contributing to the higher expression level of MYB24 in ‘TRIR-2’ than that in ‘TRIR-1’ are still unknown. More research focusing on the expression pattern and regulatory role at different leaf developmental stages is needed to explore the relationship between CcMYB24 and the formation of novel leaf traits in ‘TRIR-2’.

The content of pigments strongly influences the color of leaves, and in this study, carotenoid concentration was found to be significantly lower in transgenic Arabidopsis than in WT plants (Fig. 7A). It is known that carotenoid mutants in Arabidopsis can cause developmental defects in plastids, resulting in albino patches on leaves (Bartley & Scolnik, 1995). However, the higher concentrations of chlorophyll a, chlorophyll b, and total chlorophyll in transgenic Arabidopsis indicated that the expression of CcMYB24 was not the direct cause of the albino phenotype in ‘TRIR-2’ leaves, as chlorophylls a and b are the primary sources of green color.

Several studies have shown that MYB TFs work as regulators of primary and secondary metabolism in plants, and are essential for regulating pathways involved in plant growth, development, and responses to biotic and abiotic stress (Wang, Niu & Zheng, 2021). Plant metabolites include many types of compounds, such as sugars, proteins, flavonoids etc. Evidence has shown that the MYB TFs participate in flavonoid biosynthesis, and that the expression of some MYB transcription factor genes could promote flavonoid accumulation in plants, which could result in increased tolerance to environmental stresses (Jin et al., 2022). In the present study, accumulation of total flavonoids in transgenic Arabidopsis was significantly lower than in WT individuals (Fig. 7B), indicating that CcMYB24 may negatively influence the accumulation of total flavonoids. Some MYB genes could also influence sugar transport and promote accumulation of soluble sugars and proteins. For instance, two MYB proteins, CSA2 and CSA, were proved to regulate sugar transport in rice (Wang et al., 2021). Overexpression of MYB1 in Arabidopsis resulted in accumulation of greater concentrations of soluble sugars in transgenic individuals than in the WT plants (Cheng et al., 2013). However, the concentration of soluble sugars and proteins was not affected by overexpression of CcMYB24 in Arabidopsis (Figs. 7C and 7D), indicating that CcMYB24 may not be involved in the determination of the concentrations of soluble sugars and proteins. In addition, there were differences in traits exhibited by other plant organs between ‘TRIR-1’ and ‘TRIR-2’, such as root number and plant height (Fig. 1A). As MYB TFs play roles in many aspects of plant development (Ambawat et al., 2013; Li et al., 2019; Jiang & Rao, 2020), more research is needed to explore the function of CcMYB24 in ‘TRIR-2’.

Conclusions

In this study, a R2R3-MYB transcription factor CcMYB24 was cloned from a new foliage variant of Cymbidium, ‘TRIR-2’. ‘TRIR-2’ was an induced mutant from ‘TRIR-1’, an inter-specific Cymbidium hybrid, and it showed a series of trait variations on the leaf organ. Specifically, ‘TRIR-2’ plants produce more leaves than ‘TRIR-1’ plants. Overexpression of the CcMYB24 gene significantly increased the number of leaves in transgenic Arabidopsis plants, indicating it may play a positive role in controlling the number of leaves in ‘TRIR-2’. The number of leaves is an important trait that directly affects the quality of ornamental foliage plants, with the ornamental value of foliage plants usually being increased by increasing the number of leaves. Therefore, CcMYB24 can be considered as a potential target gene to regulate the number of leaves in molecular breeding of ornamental foliage plants.

Supplemental Information

Supplemental Information 1 The expression tag, full length cDNA and protein of CcMYB24.

Click here for additional data file.

Supplemental Information 2 Alignment of gene and protein sequence of CcMYB24 in ‘TRIR-1’ and ‘TRIR-2’.

Click here for additional data file.

Supplemental Information 3 Supplementary Tables of raw data and primers.

Click here for additional data file.

We thank Miss Songcan Yu from Yunnan Agricultural University for data analysis. We thank International Science Editing for editing this manuscript.

Additional Information and Declarations

Competing Interests

Author Contributions

DNA Deposition

Data Availability

The authors declare that they have no competing interests.

Gengyun Li conceived and designed the experiments, prepared figures and/or tables, authored or reviewed drafts of the article, and approved the final draft.

Longjie Cheng performed the experiments, analyzed the data, prepared figures and/or tables, and approved the final draft.

Zhilin Li conceived and designed the experiments, authored or reviewed drafts of the article, and approved the final draft.

Yiran Zhao performed the experiments, analyzed the data, prepared figures and/or tables, and approved the final draft.

Yuying Wang conceived and designed the experiments, authored or reviewed drafts of the article, and approved the final draft.

The following information was supplied regarding the deposition of DNA sequences:

The CcMYB24 sequence is available in the Supplemental Files and at GenBank: OQ341202.

The following information was supplied regarding data availability:

The raw data are available in the Supplemental Files.

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
