# Peer review of "Over-expression of CcMYB24, encoding a R2R3-MYB transcription factor from a high-leaf-number mutant of Cymbidium, increases the number of leaves in Arabidopsis"

_PeerJ, doi:10.7717/peerj.15490_

## Round 0.1 · original submission · Major Revisions

Your manuscript was reviewed by three independent experts in the field. All the reviewers find the work interesting but raised several issues which need to be addressed properly in a revision. The reviewers provide detailed comments in their reviews and pointed out the areas where the manuscript needs to be improved. I also read the manuscript carefully and largely agree with the reviewers’ comments.

Reviewer 1 ·

Basic reporting

no comments

Experimental design

no comments

Validity of the findings

no comments

Additional comments

I have reviewed the manuscript titled “Over-expression of CcMYB24, encoding a R2R3-MYB transcription factor from a high-leaf-number mutant of Cymbidium, increases the number of leaves in Arabidopsis.” The study has identified a MYB transcription factor, CcMYB24, from an orchid plant and functionally validated the same in Arabidopsis. Overall, the study is interesting, and well-supported the derived conclusions. I accept the manuscript in its present form.

Reviewer 2 ·

Basic reporting

Mutant in increasing leaf number is important for ornamental foliage plants, as the number of leaves is directly related to their commercial and ornamental value. In this study, a MYB transcription factor was cloned and characterized from a high-leaf-number mutant of Cymbidium. This article also described the effect of overexpression of the gene in the model plant Arabidopsis and observed the increasing of number of leaves in Arabidopsis. These findings are interesting and may be beneficial for the molecular breeding of ornamental foliage plants. Some points of this study should be carefully corrected and clarified before it can be accepted.

1. More information about the mutant ‘TRIR-2’ should be provided. I have noticed that the mutant ‘TRIR-2’ had been reported in another published paper before. Please make a short summary of that paper in INTRODUCTION.

2. Line 168: The literature of floral dip method should be cited.

3.The growth condition of Arabidopsis should be mentioned.

4. Line 189: The growth time for trait measurements of Arabidopsis should be listed.

5. In Figure 7A, 7B and Figure 6F, auxiliary lines as same as in Figure 1 should be added to the compared data to make it more comprehensible for readers. The no significance groups should also be marked.

Experimental design

.

Validity of the findings

.

Additional comments

.

Reviewer 3 ·

Basic reporting

No comment

Experimental design

No comment

Validity of the findings

Needs more supporting data

Additional comments

The overall manuscript is well written, and Author did a good job of reporting their findings, but to be accepted the manuscript needs more supportive evidence.
Author could include their RNA seq data of ccMYB24 from TRIR-1 and TRIR-2 and prove the role of MYB24 in leaf development.
Overall, brief description of trait calculation should be included in material method section. e.g., leaf length, from which point to where the leaf length is measured. Similarly, physiochemical properties of protein, Author mentions the reference for the methodology, but it will be helpful for readers, if brief description of trait calculation in current manuscript is included.
The statement, Line 212, doesn’t match with the plant picture in fig. 1A, Author wants to say leaf length or leaf area, please clarify.
Supplementary File S2 is missing in supplementary data
Fig 2B: please include high resolution image for protein alignment.
Line 278: Please rewrite.
The data shown in Figure 7A did not match with the pictures in Fig 6A, 6C. Author also tried to discuss these photosynthesis pigment, but it lacks the correct discussion. The overexpression plants look like having less chlorophyll, but the data is vice versa, please clarify.
The discussion could be improved by adding lot of supportive studies for MYB24 in Arabidopsis and other crops.

---

## Round 0.2 · accepted · Accept

I would like to appreciate authors' efforts in responding reviewers' comments and and revising the manuscript satisfactorily. The current version is suitable for publication.

Reviewer 3 ·

Basic reporting

No Comment

Experimental design

No Comment

Validity of the findings

No Comment